# The Correlation of Frequency of Work-Related Disorders with Type of Work among Polish Employees

**DOI:** 10.3390/ijerph20021624

**Published:** 2023-01-16

**Authors:** Katarzyna Kliniec, Mateusz Mendowski, Patrycja Zuziak, Mateusz Sobieski, Urszula Grata-Borkowska

**Affiliations:** Department of Family Medicine, Wroclaw Medical University, 51-141 Wroclaw, Poland

**Keywords:** work-related disorders, work in standing position, occupation, pain, ergonomics

## Abstract

Musculoskeletal disorders have a significant negative impact on the quality of life of the population. These conditions, as well as other work-related disorders, generate costs associated with treatment and work absence, which makes it a growing problem in industrialized countries. Available data from studies on individual populations of workers indicate a higher incidence of certain symptoms in these groups. Due to the lack of studies on the general population, we aimed to perform the preliminary study evaluating the occurrence of pain and work-related conditions depending on the type of occupational work among Polish employees to identify further possible areas for research. Data was collected using an electronic self-administered questionnaire, which was distributed in groups bringing together various professionals. The data obtained from 379 participants have been analyzed and divided according to performed work into sedentary, forced posture, standing, physical and requiring physical activity. Our study reveals a correlation between the frequency of work-related disorders and type of work performed in the Polish population. A significant correlation between the type of occupational work and prevalence of ankle, knee and shoulder pain, as well as heavy legs or upper limb paresthesia was found. According to our findings, female employees may be more vulnerable to lower limb symptoms. A place of residence also seems to affect the prevalence of upper back pain and heavy legs. The analysis also showed a correlation between the occurrence of hip, knee and ankle pain and the level of education of the participants. Surprisingly, lower extremity paresthesia was significantly more common among participants undertaking additional physical activity, compared to non-physically active respondents.

## 1. Introduction

Over the past decades, the type of work performed by members of highly developed societies has evolved, leading to changes in employment patterns promoting occupations associated with reduced physical activity [1,2]. A particularly significant decline in employment has occurred in economic sectors associated with monotonous work, often requiring forced body postures [3,4]. However, in recent years, there has been a dramatic increase in the number of people using electronic devices for work, which involves repetitive movements over a narrow range of motion, known as monotypic movements [1,3,4]. This type of work may be a cause of musculoskeletal disorders (MSDs), defined as injuries and dysfunction of muscles, bones, nerves, intervertebral discs, tendons and ligaments [5,6,7,8,9,10]. The most common symptom of these disorders—chronic and worsening musculoskeletal pain—is a growing problem in industrialized countries. It has a significant negative impact on the quality of life of the population and generates costs associated with attempts to treat the injured, as well as a disability pension [11,12,13,14]. Spanish authors estimated the cost associated with MSDs at over 1700 million euros in 2007, which translates into over 39 million missed workdays [15]. In the Netherlands, Germany, and the United Kingdom, one in five absenteeism from work is caused by MSDs [16]. Over 60% of respondents in a European Survey of Enterprises on New and Emerging Risks indicated musculoskeletal disorders as the most serious work-related condition [17].

Strenuous working conditions, including heavy lifting, repetitive movements, and poor posture can cause symptoms in multiple areas of body (e.g., low back, neck, upper and lower limb) [13,14,18]. These may result in progression of many MSDs such as carpal tunnel syndrome and lower back pain [19]. Although pain may also affect employees who work in a sedentary position [9,20,21,22]. Many studies show that sedentary lifestyle and lack of physical activity increase the frequency of musculoskeletal pain, which significantly reduces the quality of life [9,12,13,23]. Working in a sedentary position leads to progression of MSDs by increasing lower limb tension, impairing circulation, decreasing muscle strength and by promoting poor posture, finally causing increased pain and feelings of stiffness [6,22]. A sedentary lifestyle also raises the risk of cardiovascular disease, obesity and type 2 diabetes [24,25]. The mentioned conditions may increase the risk of lower back pain even further [12,23]. In addition, prolonged struggles with pain may lead to physical activity anxiety and thus may be a reason for deliberate limitation of exercises, which may result in the initiation of musculoskeletal diseases or exacerbation of their stage [8,20,26].

People working in jobs that require prolonged standing may experience not only lower limb or low back pain, but also discomfort, swelling and heaviness of the lower limbs and symptoms of chronic venous insufficiency [7,14,27]. The risk of lower extremity pain has been shown to increase as the percentage of prolonged standing time increases [14,20].

### Aims

As mentioned earlier, different types of work may result in the occurrence of various symptoms that are difficult to clearly identify. We aimed to evaluate the occurrence of pain and work-related conditions depending on the type of occupational work. This knowledge will allow isolating the population most vulnerable to various health complications. Due to the lack of scientific studies on the impact of the type of work on the prevalence of disease symptoms in employees, the study covered the general population (i.e., those performing various types of occupations) in order to identify the most vulnerable groups for further in-depth analysis.

## 2. Materials and Methods

In order to conduct the study, a literature review was performed; several manuscripts were found on the work-related disorders depending on the particular types of work, however, no scientific publications on the prevalence of work-related symptoms in the general population of workers were found. To adjust the design of the questionnaire used in the study, identified diseases and risk factors related to working conditions described in the scientific literature were used. Statistical data on the Polish working population were analyzed to provide a background on the study sample. The questionnaire inquiries included type of occupation, function, age, sex, height, weight, place of residence, education and additional physical activity (study questionnaire file available in Appendix A). In the form, respondents indicated whether they experienced work-related discomfort or pain from various parts of the body, along with the intensity and frequency of these complaints. The questionnaire was validated by a group of 5 working people representing different professions and they did not raise any objections to the questions included in the questionnaire. Study recruitment began on July 2021 and ended on January 2022. Responses were collected from active professionals performing different types of work among sedentary, forced posture, standing, physical work and work requiring physical activity.

The aim was to reach economically-active people who usually use social networking sites to exchange their professional experience or to look for new employment. The electronic version of the questionnaire enabled us to gain answers from various professions representatives, which allowed us to perform a cross-sectional study. To gather the data, an electronic self-administered questionnaire created by Google tools was used. The form was posted in groups created on social networking sites, bringing together representatives of various professions and residents of Poland. The respondents were invited with a short informational message, which was intended to provide basic knowledge about the conducted research, its potential scientific impact and to inform the respondent about their rights as a research participant

Members of these groups accounted for a total of 409.4 thousand people (see Figure 1). Using partial statistics available for some of the research announcements placed, we estimated that approximately 25,000 (ca. 6%) of them might have viewed the post. Inclusion criteria for the study included active work and linking the present complaints only to work conditions. We received 390 responses, giving a response rate of 1.56%. The response rate may be underestimated because it cannot be ruled out that one member may belong to many social groups. Among them 379 were enrolled in the study; 11 respondents were excluded due to incomplete submissions or filling in a questionnaire with completely contradictory data. The decision to exclude was made jointly by the three authors of the manuscript (KK, MM, PZ). 

The collected materials were statistically analyzed using Microsoft Excel (Microsoft: Redmond, WA, USA) spreadsheet (used to create the database and cross-tabulations) and Statistica software (TIBCO Software Inc., Paolo Alto, CA, USA). For measurable attributes, we calculated the arithmetic mean, median, minimum and maximum values, and standard deviation, while for qualitative ones we computed the quantitative-percentage distribution. During the statistical evaluation, the frequency of pain in specific parts of the skeleton was first summarized, from which a classification of the most common pain locations by sex and work type was created. Similar distributions were made for frequency and intensity of pain. Next, the relationship between job type and pain prevalence at a given site was analyzed. The outcome was verified using Pearson’s χ^2^ test. A significance level of 0.05 was set for all tests. The strength of the association was assessed with Cramer’s V Coefficient analysis. As the calculated value came closer to 1, the stronger the relationship was evaluated. In addition, normality tests (Shapiro-Wilk W-test) were performed to examine the differences between the average pain ratings in three different spinal segments. A nonparametric Kruskal-Wallis ANOVA test was then performed, with the overall outcome proving differences between the analyzed means.

## 3. Results

The questionnaire was completed by 390 respondents from different regions of Poland. Eleven respondents were excluded due to filling in a questionnaire with completely contradictory data (e.g., identifying oneself as not feeling pain and then rating it on the VAS scale as 10). The decision to exclude was made jointly by the three authors of the manuscript (KK, MM, PZ). Finally, 379 of them were included in the study, of which 317 were women (83.64%). Due to the electronic method of collecting data, we do not have information on how many respondents started the survey and withdrew from it. The number of participants significantly exceeded the minimum sample size to have a confidence level of 90% and margin of error of 5% in the Polish population (calculated minimum sample size was 271) [28]. The age ranged between 16 and 62 years with a mean of 30.54 years. The highest percentage of respondents were 25–34-year-old people (42.74%) with a higher education (55.67%) and living in cities with more than 100,000 residents (65.17%). Detailed sociodemographic characteristics of the study participants are shown in Table 1). 

The most common symptoms experienced by our respondents were lumbar pain (84.43%) and cervical pain (71.77%) (see Table 2). However, there was no significant difference in the occurrence of pain in these locations between the respondents in different types of work. The average pain intensity, measured using the Visual Analogue Scale (VAS), was highest for the low back (5.93 ± 1.96) and was significantly higher than in other spinal segments (see Figure 2). The strongest association between pain location and the type of work performed occurred for ankle pain (χ^2^ = 46.77, *p* < 0.01, Cramer’s V = 0.35) (see Table 3). Ankle pain was more common for work in a standing position (41.38%), physical work (40.82%) and work requiring various physical activities (36.59%) than for sedentary work (8.45%). Significantly more physical workers (49.98%) and workers whose jobs require activity (53.66%) complained of knee pain (χ^2^ = 27.06, *p* < 0.01, Cramer’s V = 0.27). Heavy legs were particularly common among physical workers (71.43%) and standing workers (65.52%) while sitting workers (42.72%) were least affected (χ^2^ = 17.49, *p* < 0.01, Cramer’s V = 0.21). Some 34.69% of physical workers reported occurrence of this symptom several times a week, while among standing workers this percentage was 31.03% (χ^2^ = 43.02, *p* < 0.01, Cramer’s V = 0.17). Upper limb paresthesia was significantly more frequent among those working in a forced position (53.19%) (χ^2^ = 12.23, *p* < 0.05, Cramer’s V = 0.18). Shoulder joint pain was less common in sedentary workers (28.64%) than in other workers (41.46% to 48.94%) (χ^2^ = 11.63, *p* < 0.05, Cramer’s V = 0.18).

We found a correlation between the sex of the workers and the occurrence of lower limb symptoms, including heavy legs, telangiectasia, varicose veins and hyperpigmentation. Feeling of heaviness was experienced by 54.89% of female and 32.26% of male participants (χ^2^ = 10.63, *p* < 0.01). Of the female respondents, 23.34% observed this symptom several times a week; however for men it was only 6.45% (χ^2^ = 13.84, *p* < 0.05, Cramer’s V = 0.19). Women also outnumbered those complaints of other lower extremity symptoms, experienced by 42.90% of female participants and 11.29% of male participants. The most common symptoms from lower limbs reported by women were telangiectasias and by men were lower limb varicose veins (χ^2^ = 43.98, *p* < 0.01, Cramer’s V = 0.34) (see Table 3).

The occurrence of thoracic spine pain and heaviness in the lower limbs correlated with the place of residence. Pain in the thoracic region of the spine was experienced by 33.33% of village residents, 56.41% residents of cities of 10,000-100,000 inhabitants and 53.04% people living in cities with population over 100,000 (χ^2^ = 8.07, *p* < 0.05, Cramer’s V = 0.15). The heaviness of the lower limbs was predominant among residents of villages (59.26%) and smaller cities (60.26%) and occurred less frequently in participants living in larger cities (46.56%) (χ^2^ = 6.09, *p* < 0.05, Cramer’s V = 0.13).

The analysis also showed a correlation between the occurrence of hip, knee and ankle pain and the level of education of the participants (χ^2^ = 9.55, *p* < 0.05, Cramer’s V = 0.16); however, the groups representing the elementary and lower secondary education comprised 4 and 3 respondents. This number of participants is too small to be considered representative and does not allow for comparison with other groups or to form a valuable conclusion.

Lower extremity paresthesias were significantly more common among participants undertaking additional physical activity (43.86%) compared to non-physically active respondents (32.79%) (χ^2^ = 13.37, *p* < 0.05, Cramer’s V = 0.19). This symptom was particularly common in individuals spending at least 6 h per week on extra activity (52.38%). A similar relationship was observed for the frequency of lower limb paresthesias among the study subjects. The occurrence of this symptom several times a week affected a higher percentage of physically active participants (6 h = 16.67%, 5 h = 13.04%, 2 h = 11.48%, 1 h = 12.00%) than inactive ones (11.32%) (χ^2^ = 62.84, *p* < 0.01, Cramer’s V = 0.17). 

## 4. Discussion

Pain complaints are one of the most significant and common problems encountered worldwide. According to a 2019 study by Statistics Poland, over half of respondents declared the incidence of pain in the four weeks preceding the survey [29]. A cross-country study conducted across 52 countries estimated that the average prevalence of pain over the 30 days prior to the interview was 27.5% [30]. One of the most common causes of pain is pain caused by MSDs, especially affecting employees [13]. Scientists evaluated the prevalence of MSDs at 1.3 billion cases globally [4]. Furthermore, some scientists highlight the fact that calculated values remain underestimated [2]. It is noteworthy that working conditions are considered one of the risk factors for MSDs; however, there have been no unequivocal studies to link individual symptoms with specific types of work [3]. It should be taken into account that active working people represent a numerous part of the population. In 2018, 3.3 billion people were in employment globally [31]. In Poland, economically active people constituted 56% of the population aged 15–89 [32]. For these reasons, the present study focuses directly on the relationship between the type of work and experienced pain. To our knowledge, this is the first study on the mentioned problem in a general population and it is not a replication of another study.

### 4.1. Target Population and Sample Comparison

The target of the study was the Polish working population. In 2020 it represented 56.5% of Polish society [33]. Unfortunately, there is no reliable data on the share of particular types of work in the Polish economy. Poland’s employment sectors, however, have been dominated by services for many years now [34]. It is a general trend observed in many European countries recently [35]. According to Eurostat, during the last 20 years, there has been a noticeable increase in service sector employment from 65% in 2000 to 73% in 2021. Working in service industries is one of the factors increasing the level of sedentary employment [36]. Our data seem to support these reports as sedentary work was the predominant type in our sample.

According to data obtained from Statistics Poland, there is noticeable sex distribution disparity between the target population of working Poles and our recruited sample (respectively 44.76% in the study and 83.64% in the sample). Since it restricts the possibility of results interpolation for a target population, we addressed this issue in the “Limitations” section in more detail. Despite the apparent disproportion in age distribution, people aged 25–44 make up over half of both the target and sample populations (52.86% and 57.25% respectively). Still, the group of workers aged 25–34 remains overrepresented in our sample, when compared to the target population. It gives the opportunity to probably diminish the negative impact of age, which promotes MSDs prevalence in the recruited sample. Furthermore, preventive measures–if applied in younger groups, starting their professional careers–may appear more effective and beneficial in the future, as their ailments may be less severe and not as consolidated as those affecting senior professionals. The percentage of people with a particular level of education is comparable in both groups. In addition, the study sample shows a slightly higher percentage of respondents living in towns and cities compared to the target population (85.75% vs. 59.62%, respectively). Comparison charts concerning sociodemographic differences between groups are shown in Figure 3, Figure 4, Figure 5, Figure 6 and Figure 7.

Our first finding is that work of any type may be associated with frequent experience of back pain, especially in lumbar and cervical regions. In a 2018 study on musculoskeletal disorders among office workers from Iran, 72.4% of the participants reported occurrence of lower back symptoms in the last 12 months [37]. Of these, 55.2% also complained of neck pain. A similar result was observed in physiotherapists whose work requires lifting and transporting patients while maintaining a forced body position [38]. Although back pain was not the most common complaint among hairdressers from India, it ranked behind knee and foot pain in this group [39].

It is noteworthy that the intensity of pain experienced by our respondents was strongest in the lower back. There are few scientific papers addressing this issue, and the results do not always support such a relationship or are inconsistent with each other. A 2017 study was focused on the correlation between sitting or standing at work and the intensity of lumbar pain [7]. Authors found no consistent correlation concerning groups of construction workers; however, prolonged sitting at work was associated with reduced lumbar pain intensity among healthcare workers. In contrast, another study involving 201 blue-collar workers showed a positive association between prolonged sitting and lower back pain intensity [40].

Several manuscripts have been related to the occupational aspect of ankle pain. Guidelines on the management of ankle sprains emphasize that high physical workload may lead to an increased risk of recurrent sprains and instability of the joint [41]. In a systematic review study, the authors noted an increased incidence of ankle pain among manual workers [42]. One of the reasons was the high physical load to which they are exposed. In our study, we also observed an increase in the incidence of ankle pain among active and manual workers. Chean et al. in a 2021 survey found that over 4455 participants came to similar conclusions [43]. However, they explained the phenomenon of widespread foot and ankle pain as a symptom often associated with the subjects’ systemic conditions (e.g., polyosteoarthritis). Another study related to this issue was conducted on a population of manual garment workers [44]. It pointed out that there was a relationship between the length of service and musculoskeletal complaints. Pain in the upper back and ankle joint were more common among workers with many years of service. The authors associate these problems with significant physical strain, movement stereotypes and working in a forced, often uncomfortable, position. Our results agree with their observations. The study conducted on a group of 636 nurses showed that foot and ankle pain is a significant problem in this population [45]. It was one of the most frequent symptoms and affected from 23 to 51% of the respondents (depending on the questionnaire used). Other authors also indicate a high prevalence of knee joint problems among nurses [46]. They emphasize that initial problems involving the ankle joint may manifest themselves as knee joint complaints. Thus, it is possible that the high prevalence of both complaints we observed in the “work that requires activity” and “physical work” groups remains related to each other, which should be further investigated.

However, this is not the only explanation we have obtained for the knee joint pain. Osteoarthritis of the knee and knee bursitis are frequently studied pathologies in terms of occupational exposure. Their high prevalence has been associated with high physical workloads as well as prolonged kneeling or squatting [47,48]. This problem was investigated in the study conducted by Kwon et al. involving approximately 4000 participants [49]. Results showed that workers in physically demanding jobs (machine operators, technicians) and workers in the agricultural industry and lower-level jobs have a significantly higher risk of osteoarthritis and chronic knee pain than office workers. They also indicate that movements that put a burden on a knee can promote pathological processes. Therefore, people working in a standing position who do not perform such movements did not show an increased risk of osteoarthritis. Knee bursitis was observed significantly more often among occupations with high physical loads and active occupations [48].

One of the risk factors for shoulder pain reported in the literature is lifting ≥10 kg above joint height. However, this association is not supported by all studies [50,51,52]. Many of them point out the greater complexity of the problem, which goes beyond occupational exposure and ergonomics [53]. A study among 803 subjects reported that spending at least 75% of work time in a sedentary position significantly increased the chances of shoulder pain resolving within 5–6 years [52]. The authors point out that sedentary workers are not exposed to lifting above the level of the joint for long periods of time, and this may be a direct cause of this result. Their observation agrees with the findings of other researchers who, in a group of 625 blue-collar workers, also noted a greater chance of pain relief in this area associated with increased sedentary work hours and reduced workloads for sedentary workers [50].

The researchers also noted a decreased likelihood of shoulder pain resolving among workers exposed to at least two factors among: manual operation of equipment, working with hands elevated above the shoulder, or using vibrating tools [54]. Their findings seem to support our observations that sedentary workers seem to be less likely to experience shoulder pain, but there is an increased risk of its occurrence among workers exposed to various stresses (physical, overactivity) during work. 

Heavy legs was a common symptom in manual and standing workers participating in our study. Very few publications address the association between this symptom and occupational activity, despite the fact that working conditions such as prolonged standing promote venous disorders [55]. The problem of prolonged standing also affects manual workers as proven by a 2019 study on the relationship between static standing time at work and the intensity of pain in the lower extremities [56]. We linked the frequent occurrence of heavy legs with the age of the participants. The feelings of heaviness, as well as other lower extremity symptoms (such as telangiectasias, varicose veins, and hyperpigmentation) that may indicate venous insufficiency, were predominant in the case of women participants. Both female sex and advanced age are risk factors for this condition [57].

Advanced age is often considered the most significant risk factor for varicose veins and chronic venous insufficiency. Higher prevalence of varicose veins in women is supported by guidelines for management of chronic venous disease [58]. However, there was no clear relationship between the incidence of telangiectasia and sex in studies included in this review [58]. Furthermore, some reviews report a higher prevalence of varicose veins among men [59]. It was also the most frequently reported symptom of venous insufficiency among the men in our study. It has to be noted that the participants in our study were dominated by women. Therefore, the statistically significant sex differences described in our study should be related to other studies with larger male representation.

Another of our findings was that place of residence may impact the incidence of heavy legs and thoracic spine pain. However, there are no studies related to this correlation. The significance of this connection and its possible causes are unknown. Therefore, further research is needed in order to explain the causes of this observed phenomenon.

Age may also be a risk factor for many others with work-related symptoms. A study investigating a population of nurses from Iran found that the prevalence of elbow and upper back symptoms increased with age [60]. Similar conclusions were drawn by Hossain et al. in the study among Bangladeshi garment workers [44]. The described relationship involved the elbow, hip, knee, and ankle joints. The incidence of shoulder and lumbar spine pain with age was also shown to increase significantly among truck drivers [61]. These analyses confirm our findings of an increased incidence of elbow, hip and shoulder pain with respondents’ age. That would stay in agreement with the well-explored theory of tissue ageing, which involves a number of cellular and metabolic changes resulting in pain and loss of mobility [62]. On the other hand, a research study conducted on Indian hairdressers showed a higher prevalence of lower and upper back complaints in the group of younger subjects. However, these employees were working longer daily hours, which undoubtedly contributed to the symptoms [39].

Several studies have shown that high Body Mass Index (BMI) is a risk factor for work-related muskuloskeletal disorders (WRMSDs) involving any area of the body. This observation was echoed with a group of loggers in the Ark-La-Tex region, also linking the occurrence of symptoms to high BMI [63]. However, researchers evaluating the prevalence of these symptoms among fishermen did not support such an association [64]. Our outcomes seem to support findings where increased body weight is related to a higher prevalence of WRMSDs. Statistical significance was found for both the elbow and hand joints in a given population. There is little literature focused on specific body areas when considering the association of WRMSDs with BMI. Further research on the correlation of excessive body weight with ailments of certain body areas seems mandatory.

The cause of the relationship between physical activity and lower extremity paresthesia among study subjects remains unclear. Some of the reasons may be additional overloading of the musculoskeletal system already strained by work, aggravation of discopathy or joint disorders [65].

Due to the multiple problems caused by MSDs, it is reasonable to take measures aimed at reducing these conditions [66]. Therefore, many employers undertake health promotion initiatives [67]. Preventive measures must consider age, sex and physical abilities of employees and, importantly, the type of work performed, taking into account for example the different nature of physical work [68]. MSDs prevention mainly involves reduction of physical load, modification of inappropriate work methods or habits, and access to information about actions offered by the company [69].

### 4.2. Confounders 

Several confounders may have influenced the conducted study. The studies carried out earlier indicated that an increased Body Mass Index (BMI) may be an independent risk factor for MSDs [70,71,72,73]. It is attributed to mechanical and metabolic factors and increased stress on joints. It requires particular emphasis since a growing number of people who are overweight and obese are reported worldwide [70]. About one in four Poles suffered from obesity in 2014 [71]. Over the past decades in Europe, the percentage of obesity and overweight cases increased from 15.5 to 22.9% and from 48 to 59.6%, respectively [70]. Currently, 60% of European adults suffer from these conditions [72]. It is estimated that about 2 billion adults are overweight globally. The regions most affected by obesity are Europe and America. It is worth mentioning that excessive weight may be associated with sedentary work [73]. Therefore, this type of work can be considered both a direct and an indirect risk factor for MSDs. 

Another confounding variable is the effect of age on the occurrence of MSDs [74]. Both European and Polish societies are aging rapidly, with a noticeable impact on health [75,76]. From 1950 to 2012, the median age of Poles increased from 28.8 to 38.5 years. Records from 2019 show that there were 703 million elderly people (65 years and over) worldwide [77]. 

Musculoskeletal aging and associated cellular and metabolic changes, such as bone loss, increase in osteoclast activity, decrease in osteoblast activity, cartilage thickness, tensile strength, chondrocyte activity and growth factor response affect the progressive degeneration of the joints and the occurrence of secondary pain [62]. However, age-related risks should not have affected our findings due to the relatively young participants of the study. The limitations section addresses this issue in more detail.

### 4.3. Limitations 

The study contains flaws as a result of imperfections in the method of obtaining respondents. The main problem of this study is its relatively small sample size, which probably limits the possibility of generalizing obtained results to the whole population. Numerous attempts were made to increase the number of participants, however the response rate remained low. The number of participants enrolled in the study limited the possibility to perform multivariate analyses, which are believed to be more accurate in bigger samples. Moreover, a low response rate may have resulted in the occurrence of a selection bias.

Another limitation, which is also related to the selection bias, is underrepresentation of men in our study. Part of the explanation for this phenomenon is the sex structure of Internet users. The number of women using social media exceeds the number of men, which was reflected in the relatively low representation of men in the research sample [78]. Additionally, women complete questionnaires and research surveys more often [79]. Another reason for the phenomenon in our study may be the difference in pain perception depending on sex. Studies show that the risk of developing painful conditions is higher in the female population [80]. All of these factors may explain why women make up the vast majority of respondents in the survey.

Furthermore, the age distribution in our sample differs from the target population. The probable cause of this effect is the electronic method of recruiting participants. However, it is noteworthy that our findings concern the relatively young population, which can be both a disadvantage and an advantage of the study. On the one hand, the surveyed sample does not accurately reflect the age structure of Polish employees, but on the other it is focusing on the younger population who already suffer from work-related ailments. This indicates areas where quick actions can be relatively effective in reducing the workload, subsequent complications, rehabilitation and treatment costs and other social burdens due to the likely longer duration of work exposure of young people.

Another limitation of our online study is the inability to verify the answers reported by participants, anthropometric parameters, or perform a physical examination; not to mention developing a proper physician-patient relationship. That is why we found it inappropriate to ask our respondents about their chronic conditions or medications. However, it must be noted that some of them, for example depression, anxiety, or anti-inflammatory drugs, may affect pain perception [81,82,83,84]. In order to collect data as faithfully as possible, a visual analog scale (VAS) was included to assess the severity of pain, along with graphic visualizations for questions concerning skin lesions [85].

Furthermore, considering the above-described ratio of responses to the number of members of social media groups where the survey was published, it is important to note the phenomenon of the self-selection of respondents. It is believed to be one of the main factors for unrepresentativeness in surveys conducted via online questionnaires [86,87]. The attitudes of study participants may have influenced their decision to take part in the survey. The theory of cognitive dissonance allows us to assume that people who use mass media search for information consistent with their beliefs and reject contradictory information [88]. Therefore, it is likely that the survey mainly included participants who consider their occupation to affect their pain [89].

## 5. Conclusions

The results of our preliminary study indicate that work-related disorders are a common problem in the studied population and the type of work performed is correlated with the incidence of specific WRMSDs and other disorders affecting lower limbs. The most frequently reported symptom in the survey was lumbar and cervical pain with the greatest intensity of pain affecting the low back. The occurrence of symptoms in specific areas of the body was frequently linked to the performance of a particular type of work. We found a strong correlation between ankle pain and work in the standing position, physical work and work that requires activity which may be due to the high physical load of these workers. Our study supports reports of other research on the correlation between physically demanding work and knee pain. According to our results, heavy legs are commonly reported by surveyed physical and standing workers. Further research is needed to explain this phenomenon. We support the findings of other studies that suggest sedentary work is associated with a lower incidence of shoulder pain. It is a complex problem; however, reduced workload is considered one of the causes. Due to medical and economical problems caused by WRMSDs, it is necessary to further investigate this issue and develop optimal preventive measures in consideration of the type of work. We have confirmed that the type of work performed influences the incidence of specific MSDs and other conditions related to lower limbs disorders. Understanding the individual health hazards at specific workplaces will pave the way to implement preventive actions and increase awareness of the need to maintain appropriate ergonomic measures. We believe that the limitations of our study may serve as guidance to project an improved program of research on this topic which will be capable of confirming our findings. We believe that the findings will allow further steps towards pain prevention among the working society.

## Figures and Tables

**Figure 1 ijerph-20-01624-f001:**
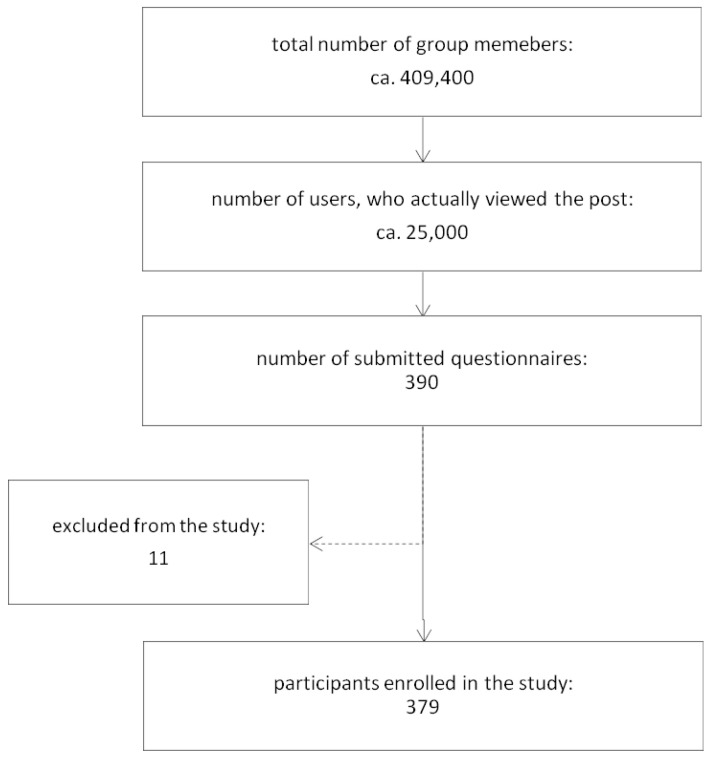
A flow chart of study participants’ recruitment strategy.

**Figure 2 ijerph-20-01624-f002:**
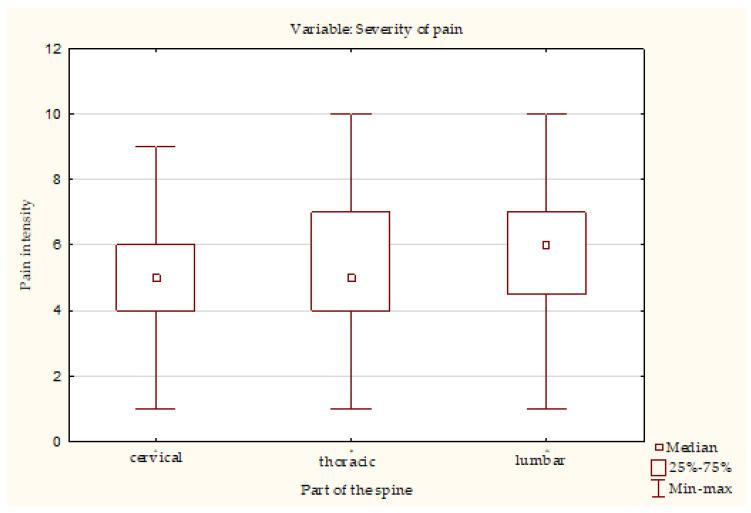
Difference in pain intensity in certain spine section.

**Figure 3 ijerph-20-01624-f003:**
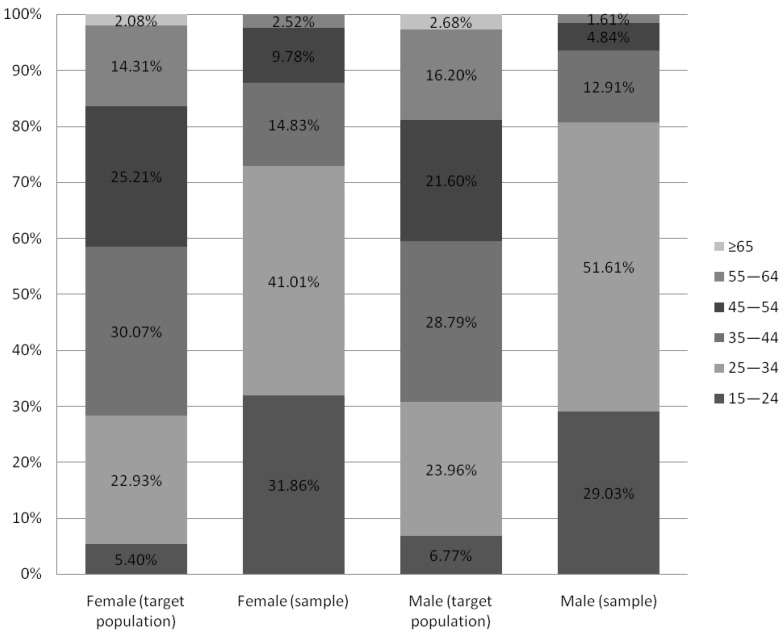
Age distribution of target population and study sample comparison (based on Statistic Poland). Note: Status report on target Polish working population in the 4th quarter of 2020 according to “Yearbook of Labour Statistics 2021”; Statistics Poland.

**Figure 4 ijerph-20-01624-f004:**
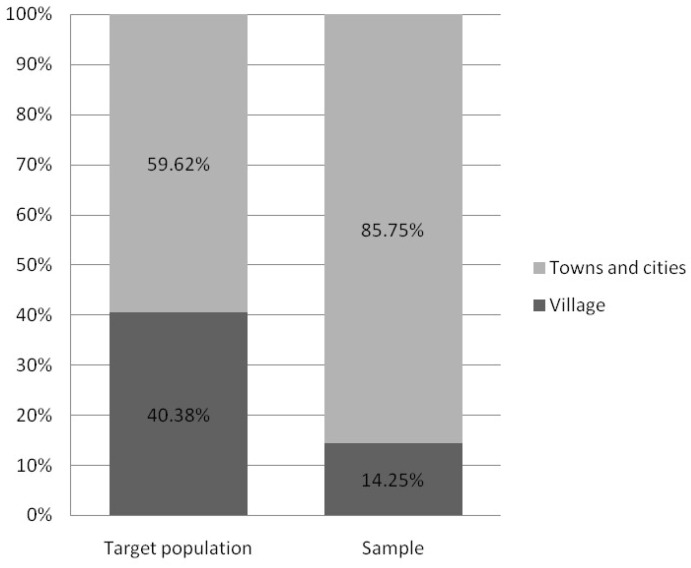
Chart presenting place of residence differences between study groups (based on Statistic Poland). Note: Status report on target Polish working population in the 4th quarter of 2020 according to “Yearbook of Labour Statistics 2021”; Statistics Poland.

**Figure 5 ijerph-20-01624-f005:**
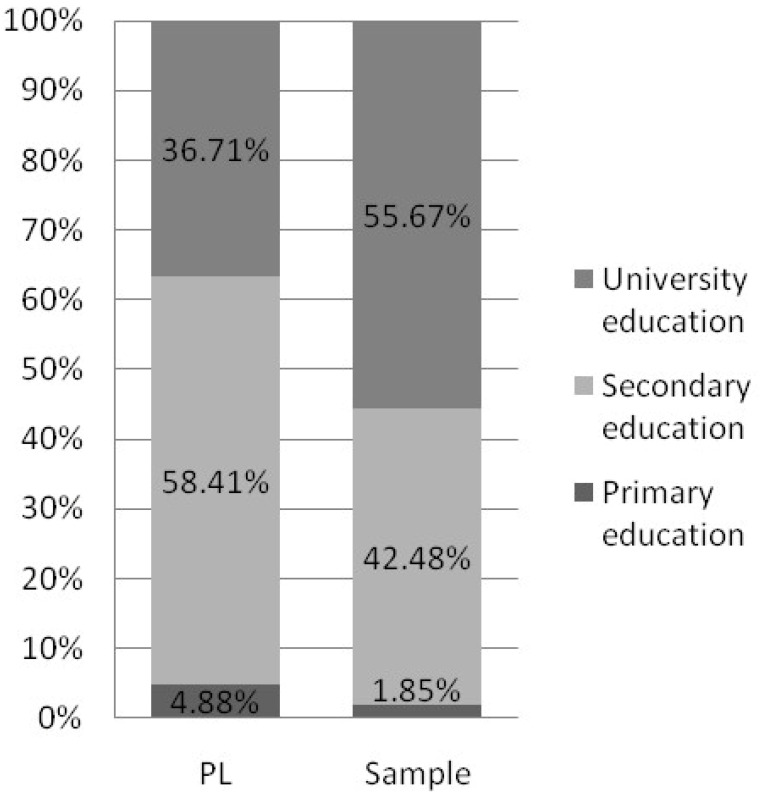
Chart presenting differences in the level of education between the study groups (based on Statistic Poland). Note: Status report on target Polish working population in the 4th of quarter 2020 according to “Yearbook of Labour Statistics 2021”; Statistics Poland.

**Figure 6 ijerph-20-01624-f006:**
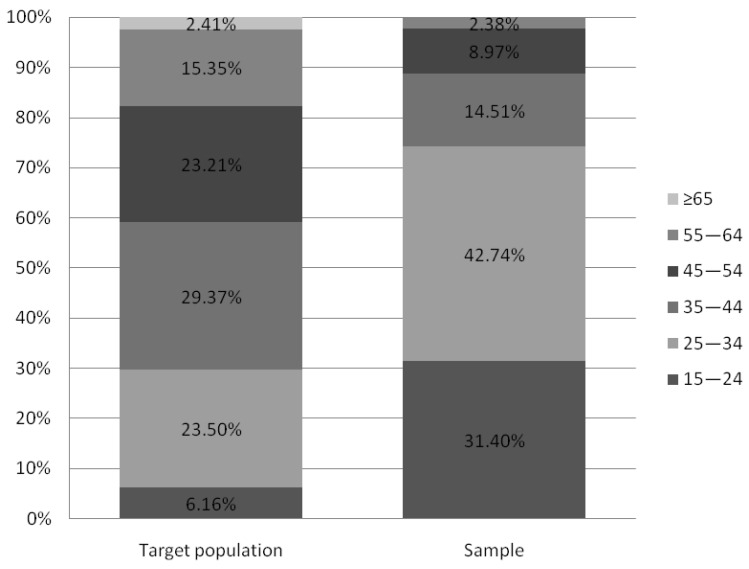
Age distribution in the target population and the study sample (based on Statistic Poland). Note. Status report on target Polish working population in the 4th quarter of 2020 according to “Yearbook of Labour Statistics 2021”; Statistics Poland.

**Figure 7 ijerph-20-01624-f007:**
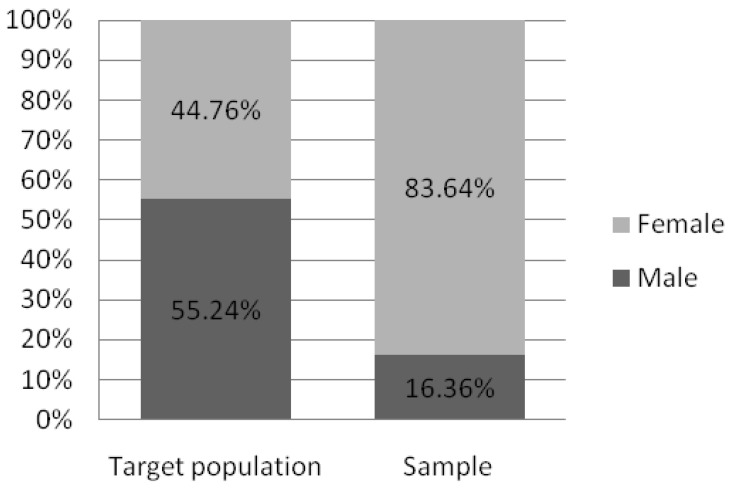
Sex distribution in the target population and the study sample (based on Statistic Poland). Note: Status report on target Polish working population in the 4th quarter of 2020 according to “Yearbook of Labour Statistics 2021”; Statistics Poland.

**Table 1 ijerph-20-01624-t001:** General information about participants.

Baseline Characteristic	Sedentary Work		Work in a Standing Position		Physical Work		Work in a Forced Position		Work That Requires Activity		All Professions	
	N = 213		N = 29		N = 49		N = 47		N = 41		N = 379	
	N	%	N	%	N	%	N	%	N	%	N	%
Sex												
Female	175	82.16	27	93.10	38	77.55	40	85.11	37	90.24	317	83.64
Male	38	17.84	2	6.90	11	22.45	7	14.89	4	9.76	62	16.36
Age												
<25 years	57	26.76	14	48.28	18	36.73	10	21.28	20	48.78	119	31.40
25–34 years	101	47.42	12	41.38	16	32.65	22	46.81	11	26.83	162	42.74
35–44 years	34	15.96	1	3.45	7	14.29	9	19.15	4	9.76	55	14.51
45–54 years	17	7.98	1	3.45	5	10.20	5	10.64	6	14.63	34	8.97
55+ years	4	1.88	1	3.45	3	6.12	1	2.13	0	0.00	9	2.37
Education												
Primary education	3	1.41	1	3.45	0	0.00	0	0.00	0	0.00	4	1.06
Lower secondary education	1	0.47	2	6.90	0	0.00	0	0.00	0	0.00	3	0.79
Secondary education	34	15.96	8	27.59	27	55.10	8	17.02	8	19.51	85	22.43
University education	134	62.91	8	27.59	18	36.73	38	80.85	13	31.71	211	55.67
University student	41	19.25	10	34.48	4	8.16	1	2.13	20	48.78	76	20.05
Place of residence												
Village	27	12.68	6	20.69	8	16.33	6	12.77	7	17.07	54	14.25
Town inhabited by 10,000–100,000 people	45	21.13	2	6.90	16	32.65	8	17.02	7	17.07	78	20.58
City more than 100,000 people	141	66.20	21	72.41	25	51.02	33	70.21	27	65.85	247	65.17

**Table 2 ijerph-20-01624-t002:** Pain frequency and intensity among respondents.

Symptoms	Work in a Standing Position				Sedentary Work				Physical Work				Work That Requires Activity				Work in a Forced Position				All Professions			
	Frequency		Intensity		Frequency		Intensity		Frequency		Intensity		Frequency		Intensity		Frequency		Intensity		Frequency		Intensity	
	n	%	$\bar{X}$	SD	n	%	$\bar{X}$	SD	n	%	$\bar{X}$	SD	n	%	$\bar{X}$	SD	n	%	$\bar{X}$	SD	n	%	$\bar{X}$	SD
Cervical pain	18	62.07	5.11	1.78	162	76.06	5.11	1.59	32	65.31	5.72	1.85	26	63.41	5.27	1.54	34	72.34	5.44	1.71	272	71.77	5.24	1.65
Thoracic pain	12	41.38	5.42	2.54	108	50.7	5.08	1.92	31	63.27	5.65	1.89	15	36.59	4.87	1.64	27	57.45	5.11	1.67	193	50.92	5.18	1.9
Lumbar pain	24	82.76	6.21	1.69	180	84.51	5.87	2.07	41	83.36	6.07	1.81	35	85.37	5.69	1.86	40	85.11	6.13	1.9	320	84.43	5.93	1.96
Shoulder pain	13	44.83	4.92	1.71	61	28.64	5.15	1.98	22	44.9	6.24	1.84	17	41.46	5.29	2.05	23	48.94	5.48	2.13	136	35.88	5.37	1.98
Elbow pain	2	6.9	6	2.83	32	15.02	5	1.76	10	20.41	5.2	1.62	3	7.32	4	0	10	21.28	4.9	1.6	57	15.04	5	1.67
Hand joints pain	13	44.83	5.15	2.3	112	52.58	4.85	1.97	29	59.18	5.24	1.75	15	36.59	4.67	2.09	27	57.45	4.78	2.03	196	51.72	4.9	1.97
Hip pain	5	17.24	5.4	1.52	54	25.35	5.21	1.97	18	36.73	5.67	2.03	12	29.27	4.67	1.83	13	27.66	5.58	2.43	102	26.91	5.28	1.99
Knee pain	7	24.14	5	1.73	46	21.6	4.72	2.1	24	49.98	5.13	2.01	22	53.66	4.32	1.59	14	29.79	5.64	2.37	113	29.82	4.86	2.01
Ankle pain	12	41.38	5.42	2.27	18	8.45	5.06	2.48	20	40.82	5.55	2.04	15	36.59	4.67	2.29	10	21.28	5.5	1.96	75	19.79	5.23	2.2
Lower extremity paraesthesia	10	34.48	-	-	90	42.25	-	-	16	32.65	-	-	20	48.78	-	-	20	42.55	-	-	156	41.16	-	-
Heavy legs	19	65.52	-	-	91	42.72	-	-	35	71.43	-	-	24	58.54	-	-	25	53.19	-	-	194	51.19	-	-
Varices	1	3.45	-	-	4	1.88	-	-	2	4.08	-	-	0	0	-	-	2	4.26	-	-	9	2.37	-	-
Telangiectasia	8	27.59	-	-	62	29.11	-	-	11	22.45	-	-	14	34.15	-	-	16	34.04	-	-	111	29.29	-	-
Discolorations	0	0	-	-	1	0.47	-	-	0	0	-	-	0	0	-	-	0	0	-	-	1	0.26	-	-
Telangiectasia and varices	0	0	-	-	15	7.04	-	-	4	8.16	-	-	0	0	-	-	2	4.26	-	-	21	5.54	-	-
Telangiectasia and discolorations	0	0	-	-	1	0.47	-	-	0	0	-	-	0	0	-	-	0	0	-	-	1	0.26	-	-

Note: $\bar{X}$-arithmetic mean, SD-standard deviation.

**Table 3 ijerph-20-01624-t003:** Association between sociodemographic factors and pain prevalence at a given site.

Symptoms	Sex			Place of Residence			Education			Physical Activity			Type of Work		
	Pearson’s χ^2^ (df = 1)	Cramer’s V	Pearson’s C	Pearson’s χ^2^ (df = 2)	Cramer’s V	Pearson’s C	Pearson’s χ^2^ (df = 4)	Cramer’s V	Pearson’s C	Pearson’s χ^2^ (df = 6)	Cramer’s V	Pearson’s C	Pearson’s χ^2^ (df = 4)	Cramer’s V	Pearson’s C
Neck pain	1.16	-	0.06	1.57	0.06	0.06	7.40	0.14	0.14	3.92	0.10	0.10	5.71	0.12	0.12
Upper-back pain	0.51	-	0.04	8.07 *	0.15	0.14	6.06	0.13	0.13	10.75	0.17	0.17	8.22	0.15	0.15
Low back pain	0.27	-	0.03	5.57	0.12	0.12	3.74	0.10	0.10	9.25	0.16	0.15	0.13	0.02	0.02
Shoulder pain	0.88	-	0.05	0.54	0.04	0.04	9.36	0.16	0.16	3.36	0.09	0.09	11.63 *	0.18	0.17
Elbow pain	0.26	-	0.03	1.39	0.06	0.06	3.32	0.09	0.09	4.53	0.11	0.11	5.95	0.13	0.12
Hand pain	0.29	-	0.03	7.71	0.14	0.14	6.68	0.13	0.13	7.02	0.14	0.13	6.09	0.13	0.13
Upper extremity paraesthesia	0.46	-	0.03	2.38	0.08	0.08	8.50	0.15	0.15	5.09	0.12	0.12	12.23 *	0.18	0.18
Hip pain	3.17	-	0.09	2.27	0.08	0.08	9.55 *	0.16	0.16	11.94	0.18	0.17	4.17	0.10	0.10
Knee pain	0.02	-	0.01	1.12	0.05	0.05	16.59 **	0.21	0.20	8.42	0.15	0.15	27.06 **	0.27	0.26
Ankle	1.30	-	0.06	4.71	0.11	0.11	24.83 **	0.26	0.25	10.51	0.17	0.16	46.77 **	0.35	0.33
Lower extremity paraesthesia	0.18	-	0.02	1.03	0.05	0.05	1.41	0.06	0.06	13.37 *	0.19	0.18	3.12	0.09	0.09
Lower extremity paraesthesia (frequency)	1.60 (df = 6)	0.07	0.06	-	-	-	-	-	-	62.84 ** (df = 36)	0.17	0.38	25.16 (df = 24)	0.13	0.25
Heavy legs	10.63 **	-	0.17	6.09 *	0.13	0.13	6.34	0.13	0.13	8.67	0.15	0.15	17.49 **	0.21	0.21
Heavy legs (frequency)	13.84 * (df = 6)	0.19	0.19	-	-	-	-	-	-	50.20 (df = 36)	0.15	0.34	43.02 ** (df = 24)	0.17	0.32
Other Lower extremity symptoms	43.98 ** (df = 6)	0.34	0.32	7.39 (df = 12)	0.10	0.14	16.06 (df = 24)	0.10	0.20	35.95 (df = 36)	0.13	0.29	16.24 (df = 24)	0.10	0.20

* *p* < 0.05. ** *p* < 0.01.

## Data Availability

The data presented in this study are openly available in FigShare at https://figshare.com/articles/dataset/data_xlsx/21507729 accessed on 3 January 2023.

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
