# Peer review of "The Correlation of Frequency of Work-Related Disorders with Type of Work among Polish Employees"

_ijerph, 2023, doi:10.3390/ijerph20021624_

Round 1
Reviewer 1 Report (Previous Reviewer 2)
Estimated Authors,
I've been invited to review a revised version of the paper "The correlation of frequency of work-related disorders with type of work among Polish employees".
Substantial improvements have been performed, and nearly all the shortcomings that appeared at the end of the previous review have been solved.
As a consequence, also my comments have been positively modified and I'm endorsing the eventual acceptance of this paper.
This manuscript is a resubmission of an earlier submission. The following is a list of the peer review reports and author responses from that submission.
Round 1
Reviewer 1 Report
This manuscript is very relevant to our current challenges of preventing chronic diseases . It is also useful in that it is conducted with participant's input not medical data.
The conclusions are a somewhat over scientific data presented and in my opinion there is a need to mentioned that further validation of the hypotheses should be pursued in a more vigorous manner. Pain for example and how to best deal with it.
What participants were excluded is not clearly defined either . Preexisting conditions that may contribute to the level of pain reported is not mentioned , Minor details in relation to the significance of the topic but it would help add to the importance of the study.
Reviewer 2 Report
Estimated Authors of the paper "The correlation of frequency of work-related disorders with type of work among Polish employees", I've read with interest your cross-sectional study performed as a web-based survey.
According to the present study, and to the estimates from a total of 379 workers from various regions of Poland, a large share of participants exhibited muscoloskeletal pain, particularly at cervical and lumbar level. Several univariate associations were identified by study Authors between some complaints and work tasks (i.e. sedentary work, work in forced position, etc.).
Unfortunately, from my point of view, the present paper cannot be accepted for the eventual publication on IJERPH for a series of reasons:
1) as accurately stressed by Authors, Musculoskeletal complaints are quite often reported in working age population: by assuming a very cautious 33% lifetime prevalence of Low Back Pain for the general population, minimum sample size would be estimated to around 350 cases for an uncertainty of 0.05, but well over 8000 for an uncertainty of 0.01. A potential way to avoid the limits associated with a limited sample size is represented by a careful selection of the study group, for example by focusing the analyses certain groups of workers with certain exposures (e.g. workers from the very same settings, or the very same economic group). On the contrary, the present study has recruited a limited sample, with a substantial over-representation of the female gender, from a broad range of various occupational settings. In other words, the reported estimates are of doubtful significance.
2) The recruitment strategy is unclear. Web based studies are reliable, no doubt about it, but Authors should provide further details on how the participants were invited, how was the study shared etc. We only are briefed that: "The form was posted in groups created on social networking sites bringing together representatives of various professions and residents of Poland" . Again, Authors suggest that the sample accounted to well below 1% of potential recipients, with even greater doubts about the representitivity of the study, and a substantial risk of self selection of participants, and the high complaint rate may be the eventual result of this self selection (i.e. among potential participants, actual ones were those having higher rate of complaints, being far more sensitive to the inquired topic).
3) The English translation of the questionnaire suggests that some topic have been inquired in a very generic way: if the questionnaire targeted the general population, a more easy to understand working definition of items should have been provided. For example: what is thoracic pain? even in healthcare workers research on MSD has often managed the topic of upper limb and cervical region as a whole stating how difficult may be to discriminate between those specific symptoms.
Moreover, several other factors affects the analyses:
1) Authors only did rely on univariate analyses, and the study could be significantly improved though an accurate multivariate analysis targeting outcome variables;
2) over-representation of female gender should be more appropriately discussed
3) stating the substantial design limits of your study, conclusions have to be specifically reshaped and reformulated avoiding any generalization
4) please discuss whether the study could benefit from being waived from an ethical committee; in fact, you're asking the respondents about their well-being, and other personal information that are also inquired by EU-GDPR.
Round 2
Reviewer 2 Report
Authors have not addressed in a consistent way my concerns, therefore also my initial judgement cannot be changed.